# E-MAPP: Efficient Multi-Agent Reinforcement Learning with Parallel Program Guidance

**Can Chang**[1,2][*], **Ni Mu**[3], **Jiajun Wu**[4], **Ling Pan**[5], **Huazhe Xu**[1,2]

[1]IIIS, Tsinghua University [2]Shanghai Qi Zhi Institute [3]Southeast University

[4]Stanford University [5]Mila, Université de Montréal
cc22@mails.tsinghua.edu.cn, xuhuazhe12@gmail.com

## Abstract

A critical challenge in multi-agent reinforcement learning (MARL) is for multiple agents to efficiently accomplish complex, long-horizon tasks. The agents often have difficulties in cooperating on common goals, dividing complex tasks, and planning through several stages to make progress. We propose to address these challenges by guiding agents with programs designed for parallelization, since programs as a representation contain rich structural and semantic information, and are widely used as abstractions for long-horizon tasks. Specifically, we introduce **E**fficient **M**ulti-**A**gent Reinforcement Learning with **P**arallel **P**rogram Guidance (E-MAPP), a novel framework that leverages parallel programs to guide multiple agents to efficiently accomplish goals that require planning over $10+$ stages. E-MAPP integrates the structural information from a parallel program, promotes the cooperative behaviors grounded in program semantics, and improves the time efficiency via a task allocator. We conduct extensive experiments on a series of challenging, long-horizon cooperative tasks in the *Overcooked* environment. Results show that E-MAPP outperforms strong baselines in terms of the completion rate, time efficiency, and zero-shot generalization ability by a large margin.

## 1 Introduction

Multi-agent reinforcement learning (MARL) has achieved significant progress by advancing the co-operation of multiple agents to accomplish complex tasks, e.g., multi-robot control [20], autonomous driving [48, 54], and video games [50, 4]. Most recent advances in MARL focus on tasks that feature behavior coordination [39] or joint motion planning [41]. However, for *long-horizon tasks* such as preparing a dish, existing methods often suffer from the inability to understand the task compositionality and subtasks' dependencies, resulting in inefficient cooperation and frequent conflicts. Therefore, a natural question to ask here is how we can solve long-horizon tasks in MARL, in the face of large state/action spaces and sparse feedback.

Long-horizon tasks are usually blessed with rich structure, and thus can be divided into a sequence of subtasks that can be resolved separately. Previous work [45, 52] has introduced programs as instructions to help a single agent understand the task hierarchy and accomplish the task. Inspired by them, we develop a general multi-agent framework, where agents can leverage programs for accomplishing long-horizon tasks together. This is a challenging problem, and three substantial issues will emerge if multiple agents are naively enforced to follow sequential programs: first, sequential programs do not explicitly express the dependencies among subtasks, thus hindering the division

---

[*]Project page: https://sites.google.com/view/e-mapp.

of jobs among agents; second, different agents might have different abilities to accomplish certain subtasks or lines of programs; third, when assigned to a subtask together, multiple agents might need to collaborate without blocking resources with each other.

As modern CPUs dispatch instructions to parallel processors, we propose a new multi-agent framework, **E**fficient **M**ulti-**A**gent Reinforcement Learning with **P**arallel **P**rogram Guidance (E-MAPP), guiding cooperation and execution by automatically inferring the structure of parallelism from programs. Specifically, we first design a domain-specific language (DSL) for multi-agent cooperation, and use multi-stage learning to ground subroutines of a given program into the agents' policy. Then, we learn feasibility functions, which entail the ability of agents to complete specific subroutines in the program in the status quo. Finally, we leverage the learned task structure to automatically enforce cooperation and division of labor among agents.

We conduct experiments on gradually more difficult challenges in the *Overcooked* [50] environment. The *Overcooked* environment requires the agents to cooperate on very long-horizon tasks, such as preparing dishes, while avoiding conflicting behaviors. Our method significantly outperforms other strong baselines in completion rates and efficiency. The program structure also enables E-MAPP to deliver superior compositional generalization to novel scenes.

Our main contributions can be summarized as follows:

- We formulate a novel task of learning multi-agent cooperation via the guidance of parallel programs.
- We present a novel framework for program grounding and long-horizon planning and instantiate the framework into practical multi-agent reinforcement learning algorithms.
- We demonstrate the effectiveness of E-MAPP in completion rates and generalization ability over existing strong baselines and provide empirical analysis in long-horizon tasks.

## 2   Related Work

**Cooperative Multi-Agent Reinforcement Learning.**   In a multi-agent cooperative game, agents collaborate with each other on a common goal [33]. Researchers have investigated many ways to facilitate agent coordination [21, 12, 53, 34, 19]. Value-based MARL algorithms engage in discovering the relationship between global value function and local value functions [46, 38], while policy-based MARL algorithms use a centralized critic [51, 26] to coordinate agent behaviors. More specifically, MAPPO [51] and MADDPG [26] leverage a fully-observable central critic to solve the issue of non-stationarity [33]. Value factorization approaches [38, 46, 37] decompose the global value function into a combination of local agent-wise utilities to cope with scalability, while policy factorization approaches [18] factorize the joint action space to coordinate marginal policies.

**Reinforcement Learning for Long-horizon Tasks.**   Reinforcement learning agents usually lack the ability to plan and reason over a long time horizon due to sparse rewards [49, 14, 40, 15, 16, 30]. Goal-conditioned reinforcement learning [36] is one of the popular paradigms to address the sparse supervision problem. Imitation learning [17, 16] is another approach to solving the sparse reward problem. Another line of work has presented automatic goal generation and selection algorithms [3, 35, 24, 9]; however, this introduces new challenges to design a suitable goal space that enjoys rich semantic meanings [13]. By contrast, our work uses the subtasks corresponding to possible subroutines in a program as goals, which are associated with domain knowledge. Hierarchical reinforcement learning [HRL;  47, 42, 28, 43, 29] is another path to solve long-horizon tasks, using a high-level policy for long-term planning and low-level policies for motion planning or specific behaviors. While our work is related to multi-agent hierarchical reinforcement learning [27], which explicitly provides the directed acyclic task graph and the necessity of cooperation of each subtask, we focus on learning task structure such as subtask dependencies, loops, and branchings from the program and judging the necessity of cooperation without additional information.

**Instruction-Guided Agents.**   Many recent advances have testified the advantages of leveraging structured prior knowledge such as task graphs [1, 22], natural languages [2, 6, 23] and programs [45, 52] to promote efficient policy learning. In contrast to other structured priors, programs stand out because of their strictly formatted and composable subroutines [5, 7]. Previous works leverage programs to enable a single agent to learn complex tasks by following programs [45, 52]. However, a plethora of new challenges have been introduced, including task dependencies, collaboration schemes,

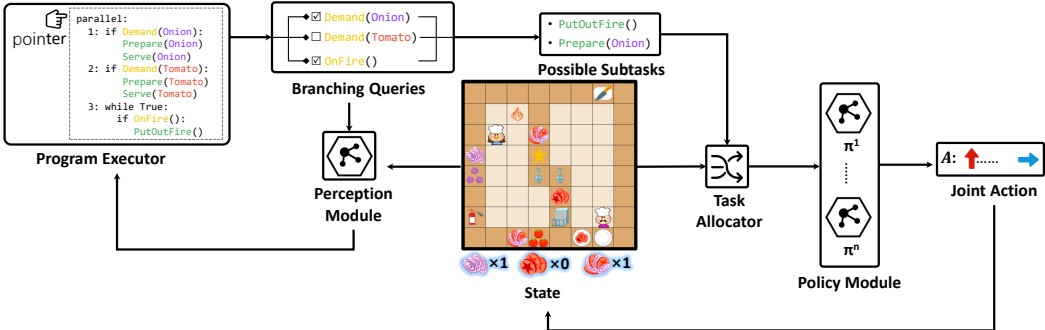

Figure 1: **The overall framework of E-MAPP.** E-MAPP includes four components: 1) A perception module that maps a query $q$ and the current state $s$ to boolean responses. 2) A program executor that maintains a pool of possible subtasks and updates them according to the perceptive results. 3) A task allocator that chooses proper subtasks from the subtask pool and assigns those to agents. 4) A policy module that instructs agents in taking actions to accomplish specific subtasks.

and others; hence, it is difficult to trivially extend the existing works in the MARL settings. To enable multi-thread orderless policy execution, we leverage "plug-and-play" auxiliary functions to infer the relationships among subtasks.

## 3 Problem Statement

### 3.1 Program Guided Cooperative Markov Game

An infinite-horizon Markov Game is defined by a tuple $(\mathcal{N}, \mathcal{S}, \mathcal{A}, \mathcal{T}, \mathcal{R}, \gamma)$, where $\mathcal{N} = \{1, \ldots, N\}$ denotes the set of $N$ engaging agents, $\mathcal{S}$ denotes the state space, $\mathcal{A} = \mathcal{A}^1 \times \cdots \times \mathcal{A}^N$ denotes the Cartesian product of all the $N$ agents' action space, $\mathcal{T} \colon \mathcal{S} \times \mathcal{A} \to \mathcal{S}$ denotes the state transition function from current state $s$ to the next state $s'$ for the joint action $a = (a^1, \ldots, a^N)$, $\mathcal{R} = \mathcal{R}^1 \times \cdots \times \mathcal{R}^N$ denotes the Cartesian product of all the $N$ agents' reward functions, where each $\mathcal{R}^i$ determines the immediate reward for the $i$-th agent from the current state $s$ and joint action $a$, and $\gamma$ is the discount factor.

At each time $t$, each agent $i$ observes the current state $s_t$, makes the decision $a_t^i$, and receives the reward $r_t^i$. In a cooperative game, the collective goal is to optimize the joint policy $\pi \colon \mathcal{S} \to A$ to maximize the sum of each agent's expected cumulative discounted rewards $\mathbb{E}_{\mathbf{a}_t^i \sim \pi^i(\cdot|\mathbf{s}_t), \mathbf{s}_t \sim \mathcal{P}} \left[ \sum_{t=1}^{\infty} \sum_{i=1}^{N} \gamma^t r_t^i \left( \mathbf{s}_t, \mathbf{a}_t^i \right) \right]$.

A program-guided Markov game is a Markov game with a factorized state space. Specifically, state $\mathcal{S} = \mathcal{S}_e \times \mathcal{S}_p$, where $\mathcal{S}_e$ is the common state space of the environment and $\mathcal{S}_p$ is the multi-pointer program space (see Section 3.2). Accordingly, the state transition function takes as input the current compounded states $(s_e, s_p)$ and joint actions $a$ and then returns the next joint state $(s_e', s_p')$. In this study, the transition function of the program space is based on predefined rules.

### 3.2 Parallel Programs

The program space consists of three components: a domain-specific language (DSL) that contains all the possible subroutines, a set of pointers that point to relevant subroutines, and a control flow that manages the pointers. A complete specification of the DSL used by our framework is in the Appendix A.1. Inside this DSL, a subroutine is a minimal executable unit in the program that corresponds to a subtask in the domain (e.g., *Chop(Tomato)*). As in previous works [45, 52], we consider two types of subroutines: perception primitives (e.g., *IsOnFire()*), which query about the status of the environment; and behavior primitives (e.g., *Chop(Tomato)*, which issue an instruction. Control flow involves branching statements (*if/else*), loops (*for/while*), and **parallelism indicators** (*repeat/parallel*). The parallelism indicators are designed for multi-thread execution. The subroutines in a *parallel* block are possible but not guaranteed to run concurrently, i.e., the agents should reason about what subroutines in a *parallel* block can be executed concurrently. The

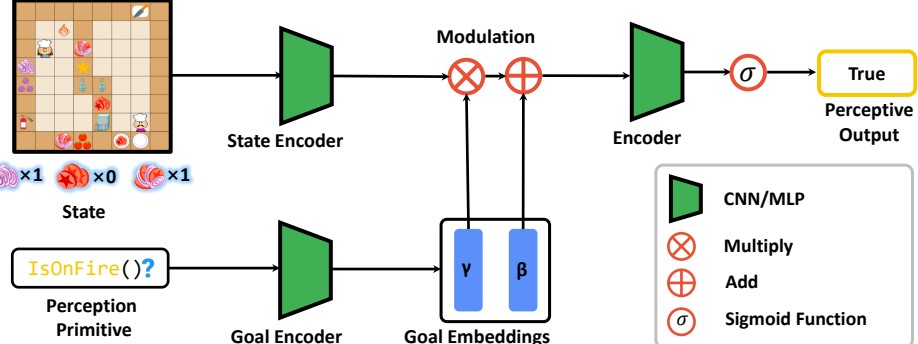

Figure 2: **The perception module**. The perception module encodes the perception primitive into modulation parameters, which operate on the state embeddings to get the goal-conditioned embeddings. These embeddings are then fed into another encoder, followed by a sigmoid function. The output is a real number in $[0, 1]$, and will be binarized for branching selection by the program executor.

subroutine in a *repeat* block can be executed many times simultaneously. We summarize all the current subroutines that are possible and executable as the *Possible Subroutine Set*. The set of pointers points to all the subroutines in this set.

### 3.3 Multi-Agent RL with Parallel Programs

In our study, we aim to develop a framework for optimizing the joint policy in a parallel program-guided cooperative Markov game. To this end, the agents must keep track of the pointers in the program, learn to reason the primitives to choose the right branches, and perform the action either collaboratively or individually to pursue high efficiency.

## 4 Method

Our goal is to enable multiple agents to cooperate to solve long-horizon tasks guided by parallel programs. There are three important factors we should consider: first, the agents should learn subtask-conditioned policies that can be composed together to accomplish long-horizon tasks and further generalize compositionally to unseen tasks; second, the agents should reason about task dependencies so that they can automatically parallelize tasks among them; and third, the agents should distinguish between cooperative tasks and non-cooperative tasks that can be achieved by a single agent to avoid competition for common resources.

We propose **E**fficient **M**ulti-**A**gent Reinforcement Learning with **P**arallel **P**rograms (E-MAPP), a multi-agent reinforcement learning framework where agents can cooperate to solve long-horizon tasks by following program guidance. As shown in Figure 1, E-MAPP includes four components: a perception module that is able to judge whether the queried status exists in the state; a parallel program executor that keeps track of subroutines and updates them based on the perception module output; a task allocator, which extracts queries of interest and obtains the corresponding set of possible subtasks from the program executor, and assigns tasks to agents; and a multi-agent policy module, which agents use to make their decisions based on the input subtask. The following of this section will introduce these four key components of E-MAPP. The complete algorithm is shown in Appendix A.3.

### 4.1 Parallel Program Executor

The program executor keeps a set of pointers pointing to possible subroutines in a domain-specific *Possible Subroutine Set*. There are four types of control flows: *if*-routine, *while*-routine, *parallel*-routine, and *repeat*-routine. Meanwhile, there are two types of subroutines in our program: behavior primitives and perception primitives.

**Control flows.** An *if*-routine contains a condition statement (usually a perception primitive) and subroutines in the corresponding blocks. A *while*-routine contains a condition statement and a looping

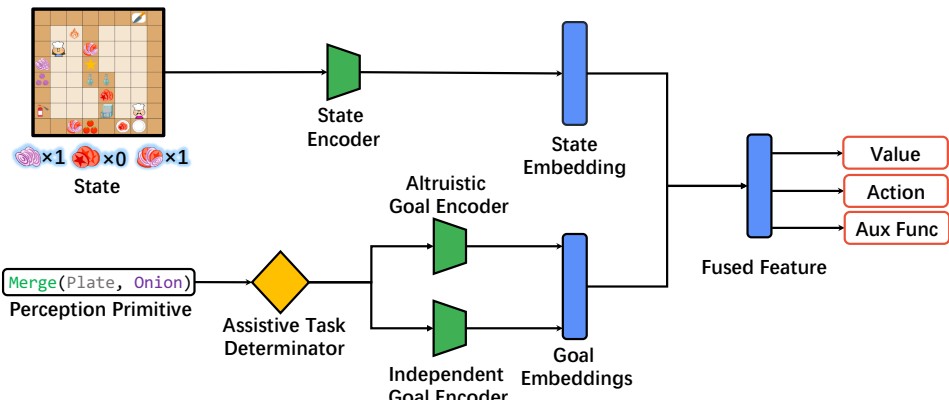

Figure 3: **The policy module**. We adopt a goal-conditioned reinforcement learning framework to determine the policy. We encode goals with a goal encoder and fuse them into the state features. If an agent is assistive, the goal is encoded with an altruistic goal encoder instead. The network has three outputs, including the value function, the action distribution, and the auxiliary functions.

block of subroutines. A *parallel*-routine contains parallel blocks of subroutines that are possible to be executed simultaneously. A *repeat*-routine contains an unconditioned block of subroutines that can be executed multiple times by different agents.

**Subroutines.** A behavior primitive (e.g., *Chop(Tomato)*) corresponds to a subtask that must be completed by the agents. As for a perception primitive (e.g., *IsOnFire()*), it is a query that requires a boolean response.

After an action is performed or a response to a perceptive query is received, the program executor updates its pointers and the Possible Subroutine Set. Detailed updating rules are shown in Appendix A.2. We note that if an achieved subtask does not correspond to any subroutine in the Possible Subroutine Set, the program executor will terminate the program immediately. This guarantees that no exceptions will occur due to violating the instructed order of subtasks.

### 4.2 Perception Module

The perception module learns to map a query $q$ and the current state $s_t$ to a boolean answer $h = \phi(q, s_t)$. For example, when a perception primitive *IsOnFire()* is passed to the perception module, it learns to check the existence of fire in the environment and returns true/false. The architecture is shown in Figure 2. Specifically, we randomly sample states, queries, and the ground-truth perception $h_{gt}$ as the training dataset and train the network $\phi$ in a supervised manner. We use binary cross entropy (BCE) loss $\mathcal{L}_{\text{perception}} = BCELoss(h_{\text{pred}}, h_{\text{gt}})$ as the objective, where $h_{\text{pred}}$ denotes the perception output. In terms of architecture, we encode the perception primitive $s_p$ and the common state $s_e$ with a neural encoder. The encoded primitives are represented as $\gamma$ and $\beta$ to modulate the encoded common state. At the last layer, we use a sigmoid function to obtain the binary output. Training details and detailed architecture descriptions can be found in Appendix A.5. In this way, this module can determine whether the queried primitive exists in the state to aid the agents' decision-making.

### 4.3 Policy Module

The policy module grounds agents' actions with the programs and encourages cooperation in completing the tasks. The overall policy learning procedure advocates a subtask-conditioned reinforcement learning framework as shown in Figure 3. Our algorithm backbone is MAPPO [51] and consists of two stages. More concretely, we first learn a policy for each agent where other agents' policies are fixed. The input is the state and the encoded subtask. The reward signal is based on whether a subroutine is executed correctly. Then we enable multiple agents to coordinate by learning a joint policy for accomplishing collective goals cooperatively and efficiently, with the help of auxiliary functions and a task allocator. We use the same architecture to fuse state and goal features as that in the perception module in Figure 2. We also leverage self-imitation learning to tackle the challenge of sparse rewards, which is shown in Appendix A.4

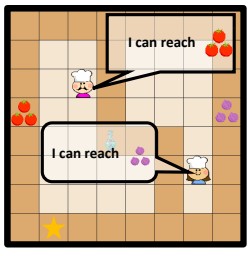
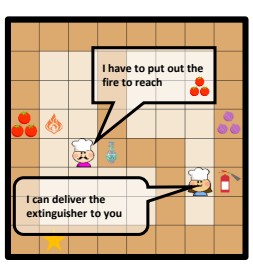
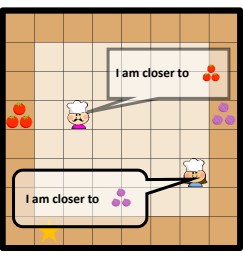

(a) reachability            (b) feasibility            (c) cost-to-go

Figure 4: **The auxiliary functions**. Figure 4a shows the reachability that indicates if an agent can reach an object. Figure 4b shows the feasibility that indicates whether a subtask can be achieved by the agents collectively. Figure 4c shows the cost-to-go function that indicates time consumption.

**Learning to cooperate on a subtask.** After obtaining single-agent policies, we then encourage agents to accomplish harder tasks that require cooperation. When assigned a cooperative subtask, one of the agents is in charge of finalizing the task. The rest of the agents are assistive. For example, the assistive agents might pass an onion to the leading agents to chop. In practice, we randomly appoint one agent as the leading agent and the others as the assistive agents. The leading agent is rewarded if this subtask is completed. The reward function for the assistive agents is calculated based on the *reachability* improvement of the leading agent. We defer the formal definition of this reward function to Section 4.4. Such reward function encourages the assistive agents to help the leading agents to obtain rewards. Then, we use a MAPPO-style algorithm to obtain a cooperation policy. We also encode the leading agent's goal into the observation space of the assistive agents to enhance information sharing.

### 4.4 Task Allocator

The task allocator assigns subtasks to each agent to accomplish a long-horizon task together. Specifically, we design the task allocator for efficient coordination based on the following principles: 1) It only assigns subtasks that are feasible for the agent(s) without additional prerequisite subtasks. 2) If a subtask is cooperative, the task allocator would assign the subtask to a number of agents. 3) It assigns a subtask to the agent that has the lowest cost in terms of execution time.

To facilitate the task allocation, we propose to learn three auxiliary functions: a *reachability function*, a *feasibility function*, and a *cost-to-go function* as illustrated in Figure 4. Training details of the auxiliary functions are in Appendix A.6

**Reachability.** The reachability function $f_{reach} = f_{reach}(s_t, i, \tau)$ is defined as a boolean value to indicate if agent $i$ can complete task $\tau$ alone at the state $s_t$. For a cooperative subtask $\tau$ and a selected leading agent $i$, an extrinsic reward $f_{reach}(s'_t, i, \tau) - f_{reach}(s_t, i, \tau)$ is provided to the assistive agents for learning altruistic behaviors. To train this reachability function, we randomly sample triplets $(s_t, i, \tau)$ and obtain the ground-truth value $f_{reach}^{gt}$ through running the pre-trained non-cooperative policy. Then we optimize the network by minimizing the binary cross entropy $\text{BCE}(f_{reach}^{pred}, f_{reach}^{gt})$.

**Feasibility.** The feasibility function $f_{feas} = f_{feas}(s_t, i, \tau)$ is defined as a boolean variable indicating if an agent $i$ can complete a subtask $\tau$ with others' assistance at the state $s_t$. The training procedure of the feasibility function is similar to that of the reachability function, except that we leverage the cooperative policy instead of the single agent policy to collect training data.

**Cost-to-go.** The cost-to-go function $f_{cost} = f_{cost\text{-}to\text{-}go}(s_t, i, \tau)$ denotes the remaining timesteps for agent $i$ to accomplish the subtask from the state $s_t$. We leverage trajectories generated by a pre-trained intermediate cooperative policy from E-MAPP to train this cost function. Specifically, we randomly sample triplets $(s_t, i, \tau)$ and execute the pre-trained cooperative policy in Section 4.3

to obtain the ground-truth timesteps $f^{gt}_{cost\text{-}to\text{-}go}$ to complete the subtask $\tau$. Then we optimize the network by reducing the Mean Squared Error (MSE) $\mathcal{L}_{cost\text{-}to\text{-}go} = \text{MSE}(f^{pred}_{cost\text{-}to\text{-}go}, f^{gt}_{cost\text{-}to\text{-}go})$.

**Criteria for subtask allocation.** For a specific Possible Subroutine Set $\{\tau_1, \ldots, \tau_m\}$, we denote a legal subtask allocation as $\{T_1, T_2, \ldots, T_m\}$ such that

- $T_i$ is a list of $n_i$ agents where $n_i = |T_i|$. The agents are those in the agent set $\{1, \ldots, N\}$ who aim at completing the subtask $\tau_i$.
- For all $i \in \{1, \ldots, m\}$, if $n_i$ is larger than one, then the agent $T_i^1$ is selected as the leading agent and the agent(s) $T_i^2, \ldots, T_i^{n_i}$ are selected as the assistive agent(s).
- For $1 \le i < j \le m$, we have that $T_i \cap T_j = \emptyset$, which means no agent is assigned two subtasks simultaneously.

We compute the cost of each possible subtask allocation $\{T_1, T_2, \ldots, T_m\}$ as the sum of three terms

$$c_{\text{total}} = c_{\text{feas}} + c_{\text{cost-to-go}} + c_{\text{reach}} \tag{1}$$

,where

$$c_{\text{feas}} = \sum_{i=1}^{m} -n_i w_{\text{feas}} \log f_{\text{feas}}(s, T_i[1], \tau_i) \tag{2}$$

$$c_{\text{cost-to-go}} = \sum_{i=1}^{m} -n_i w_{\text{cost}} f_{\text{cost}}(s, T_i[1], \tau_i) \tag{3}$$

$$c_{\text{reach}} = \sum_{i=1}^{m} \mathbb{I}[n_i = 1] w_{\text{reach}} \log f_{\text{reach}}(s, T_i[1], \tau_i) \tag{4}$$

$w_{feas}$, $w_{reach}$, and $w_{cost}$ are tunable hyperparameters and $\mathbb{I}[\cdot]$ is the indicator function. The *feasibility* term encourages the agents to choose the feasible subtasks, the *cost-to-go* term encourages the agents to choose the less costly subtasks, and the *reachability* term guarantees that the subtask assigned to only one agent is non-cooperative. The *log* operator operating on a value approximating 1 will induce a huge cost, thus preventing the allocation of an infeasible or unreachable subtask to an agent. We search from the possible allocations $\{T_1, T_2, \ldots, T_m\}$ and apply the one with minimal cost as the final allocation.

In practice, we also use two hyperparameters: $c_r$ to be subtracted from the cost-to-go function to encourage the agents to finish their ongoing subtasks, and $c_i$ to be added to the cost-to-go function to avoid allocating subtasks to an agent who can never accomplish a task within a timeout threshold $t_o$.

## 4.5 Complexity Analysis

In an environment with $M$ subtasks and $N$ agents, the brute force search for an optimal allocation indeed has a complexity of $O(M^N)$. However, the practical complexity is much smaller than it. The reasons are as follows:

1. In a certain stage of a long-horizon task, only a small amount of subtasks are feasible. Thus, the subtask amount $M$ can be pruned into a smaller number $L$ by checking the feasibility function $O(M \times N)$ times and ignoring the subtasks whose feasibility functions are less than a given threshold.

2. The engaging $N$ agents can be classified into $C$ roles. The agents sharing the same role have the same reachability functions. $C$ is often a property of the task that does not scale with $N$. For example, in the overcooked environment, $C$ can be the number of connected components of the map. Note that, in E-MAPP, the assistive agents aim to increase the reachability of the leading agents. We define $C \times L$ new subtasks $(\tau, c)$, where $\tau$ comes from the $L$ feasible subtasks and $c$ comes from the $C$ roles. The goal of each new subtask $(\tau, c)$ is to help agents with role $c$ to gain reachability on subtask $\tau$. We can obtain a new subtask set of size $O(C \times L)$ by extending the original subtask set with these newly defined subtasks. Assume that the number of agents is smaller than the number of feasible subtasks (otherwise, idle agents will inevitably emerge). Under this assumption, each agent will choose to either complete a subtask alone or assist a certain group of agents with the same role, and each subtask in the new subtask set is allocated to at most one agent to avoid

conflict. Then the task allocation problem turns into finding the best matching of $N$ agents and $O(C \times L)$ subtasks with the smallest total cost, which can be solved by the Hungarian algorithm. The computational complexity is $O((N + CL)^3) \leq O((N + CM)^3)$ that scales well.

# 5 Experiments

In this section, we aim to investigate the following key questions. First, is the program guidance helpful for agents to understand long-horizon tasks in comparison with other structured information guidance? Second, are the parallel structures in the programs bring forth cooperative behaviors among the agents? Third, does the task allocator improve the completion rates and the time efficiency of the long-horizon tasks by virtue of the auxiliary functions?

## 5.1 Environment Description

To evaluate the proposed framework, we adapted the previous environment [50]mimicking the video game to a more challenging one "Overcooked v2". Concretely, we extend temporally for the horizon length of a task by adding additional behaviors such as "wash dishes" and "put out fire". As shown in Figure 5, agents can navigate through the grid world, interact with objects (e.g., tomatoes, knives), and deliver the dishes to the customers (the yellow star). The goal of the agents is to serve dishes according to the given recipes that can be divided into subtasks. More details about the environment can be found in Appendix A.7.

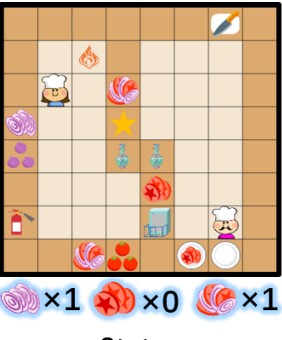

Figure 5: A sample from the *Overcooked* environment.

## 5.2 Setup

**Tasks.** We test the agents on a variety of tasks with different difficulty levels. The **easy** tasks contain only one subroutine for verification. The **medium** tasks contain two or three subroutines. The **hard** task requires the agents to complete multiple dishes that require at least two subroutines each while putting out the randomly appearing fire. We conduct evaluation in two different patterns, based on seen or unseen tasks. Note that both evaluation scheme are conducted on novel maps, which makes the task more challenging. The unseen tasks share the subroutines with the seen tasks but is never used as a goal during training. The unseen tasks are used to test whether the methods can generalize compositionally. We also note that for the tasks with a *repeat* command (e.g., repeatedly pick an onion from the supply), a targeted repeat number is preset.

**Metrics.** We use the completion rates and the average scores as the metrics. The completion rate is defined as the percentage of tasks that can be completed in an episode. The average score is the discounted cumulative rewards in an episode across all testing tasks. Agents will receive a reward of 1 when the final goal is achieved and receive a reward of 0.2 when a correct subtask is completed. Each algorithm is tested in 1000 environments and we report their average in the tables.

**Baselines.** We compare our model with two baselines: MAPPO [51] and a natural language-guided agent. Detailed descriptions are in Appendix A.9.

## 5.3 Results

**Results with seen tasks in novel maps.** Table 1 shows the completion rate and average scores on seen tasks in novel maps, demonstrating that E-MAPP excels at understanding the structure of complex tasks. As expected, the end-to-end MAPPO model performs well on short-horizon tasks, but suffers from a significant performance drop when the task becomes complex. The natural language–guided model also has a large performance drop when the horizon of the tasks becomes longer. We attribute this to the agents' failure to understand the complex task structure described in natural language without explicit structure. By contrast, E-MAPP performs well even when the tasks have a long horizon with various accidental events happening.

Table 1: **Results on the seen tasks in novel maps.** Completion rates and average scores of seen tasks in the *Overcooked* environment with three difficulty levels.

| Methods | Completion Rates | | | Average Scores | | |
|---|---|---|---|---|---|---|
| | Easy | Medium | Hard | Easy | Medium | Hard |
| E-MAPP (ours) | 98.0% | **97.5**% | **56.3**% | 1.06±0.17 | **1.12**±0.22 | **1.58**±0.60 |
| Natural language guidance | 81.9% | 48.1% | 1.01% | 0.87±0.43 | 0.63±0.51 | 0.82 ± 0.31 |
| MAPPO [51] | **100.0**% | 65.7% | 0.00% | **1.11**±0.04 | 0.79±0.31 | 0.59± 0.27 |

**Generalization with unseen tasks in novel maps.** Table 2 shows the results on the zero-shot generalization scheme. The scenarios and tasks are very different from the training domain, posing an extra challenge to all the algorithms, including the auxiliary functions of E-MAPP. We find that E-MAPP is significantly better than both the previous multi-agent RL algorithms and a language-guided agent. We attribute the success of E-MAPP to the fact that programs have better compositionality and less ambiguity. In the training time, E-MAPP learns certain subroutines from other tasks; it applies the behavior compositionally to achieve goals never seen during training. More visualized results are in Appendix A.8.

Table 2: **Results on the unseen tasks in novel maps.** Completion rates and average scores of unseen tasks with two difficulty levels.

| Methods | Completion Rates | | Average Scores | |
|---|---|---|---|---|
| | Medium | Hard | Medium | Hard |
| E-MAPP (ours) | **100.0%** | **43.7%** | **1.13** ± 0.10 | **0.99** ± 0.22 |
| Natural language–guided model | 58.4% | 0.0% | 0.58 ± 0.50 | 0.48 ± 0.21 |

**Ablation of auxiliary functions.** We analyze the significance of the proposed key components in our model by comparing our model with the variants that removes 1) the feasibility predictor 2) the reachability predictor 3) the cost-to-go predictor. The completion rates and the average scores are shown in Table 3. We find that all the components are important to E-MAPP. Specifically, we find that removing the reachability predictor from E-MAPP leads to a significantly increased average timestep, and this is the component that provides the largest performance gain. We attribute this to the reachability function that pointed out the subtasks that can be completed individually, providing an impetus for agents to parallelize.

**Ablation of parallelism in the programs.** We compare the proposed parallel programs with sequential programs in the same environments. In Table 3, we find that the agents with parallel programs use 15% fewer time steps to accomplish the same goal as those with sequential programs. These results show that the parallel programs are central to E-MAPP in terms of time efficiency.

Table 3: **Ablation study.** Completion rates, average scores, and timesteps when one of the key components of E-MAPP is removed.

| | Completion Rates ↑ | Average Scores ↑ | Average Timesteps ↓ |
|---|---|---|---|
| E-MAPP | **56.3%** | **1.58** ± 0.60 | **17.6** |
| w/o feasibility predictor | 38.5% | 1.42 ± 0.56 | 21.24 |
| w/o reachability predictor | 43.8% | 1.45 ± 0.50 | 23.37 |
| w/o cost-to-go predictor | 52.0% | 1.46 ± 0.45 | 20.42 |
| sequential program | 48.8% | 1.51 ± 0.47 | 20.33 |

**Partially observable environments.** We conduct an experiment to show that E-MAPP can still outperform other methods in a partially observable environment. In the new setting, the observation of each agent is only part of the map within reach. Table 4 shows the results. Under the new setting, E-MAPP can still learn to allocate sub-tasks to agents and accomplish tasks efficiently.

Table 4: **Additional experiment in partially observable environments** we conduct an experiment to show that E-MAPP can still outperform other methods in a partially observable environment.

| model | score | completion rate |
|---|---|---|
| E-Mapp(partial obs) | 1.01±0.38 | 27.1% |
| E-MAPP(original) | 1.58±0.60 | 56.3% |
| MAPPO | 0.59± 0.27 | 0.0% |

## 5.4 Visualization of Learned Behaviors

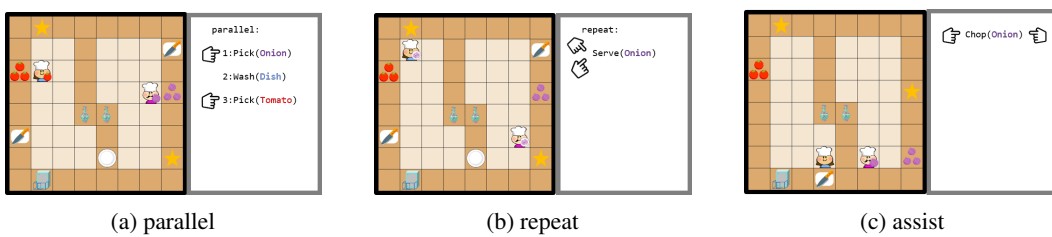

|           (a) parallel           |           (b) repeat           |           (c) assist           |

Figure 6: **Learned behaviors of E-MAPP**. Figure 6a shows that two agents are assigned two subtasks concurrently. Figure 6b shows that both agents are trying to repeatedly perform the same task. Figure 6c shows that one agent is passing an onion to the agent who can chop it.

In Figure 6, we show the representative learned behaviors of E-MAPP. In Figure 6(a), the two subroutines are allocated to two agents separately. This demonstrates the effectiveness of the *feasibility* function, with which the task allocator successfully identified the parallelizable subtasks that can be executed concurrently. We note that the *parallel* indicator in the program only suggests the agent trying to identify what subtasks are parallelizable rather than providing a ground-truth task structure. In Figure 6(b), we find that the two agents are performing the same task in a *repeat* subroutine independently. We show the allocator can assign tasks to multiple agents that are reachable to the goal. In Figure 6(c), we show that an assistive agent is passing an onion to the leading agent to chop. This shows how the agents learn to cooperate with each other. More visualizations can be found in Appendix A.8.

## 5.5 Scalability to New Domains

We also investigate whether E-MAPP can be effective on tasks with continuous action space based on the *Stacking* environment. More details and demos in *Stacking* can be found in Appendix A.10. We discuss some potential parallel-program synthesis approaches when applying E-MAPP to a new domain in Appendix A.14

## 6 Conclusion

In this paper, we first formulate the problem of program-guided multi-agent tasks. We propose Efficient Multi-Agent Reinforcement Learning with Parallel Programs (E-MAPP), an effective framework that uses a type of parallelism-aware program for multi-agent collaboration and a task allocation strategy via a set of learnable auxiliary functions. The results show that our algorithm can infer the task structure and significantly boost completion rates, efficiency, and generalization ability on long-horizon tasks.

**Limitations.** Our current framework does not consider more challenging scenarios such as dynamic scenes or generalization to novel objects. We believe that this work opens a welcoming avenue to this research direction, and more future works will address additional challenges.

**Acknowledgements.** We thank Yuping Luo and Zhecheng Yuan for their careful proofreading and writing suggestions.

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
