# A Appendix

## A.1 Domain Specific Language (DSL) Specifications

Table 5 shows the domain-specific language (DSL) designed for E-MAPP in the *Overcooked-v2* environment.

Table 5: **DSL for the Overcooked-v2 Environment.** We list types of constituent elements of the DSL. We provide the context-free grammar for generating all legal programs.

| Type | Instances |
|---|---|
| Program p | `def main():s` |
| Item t | `FreshOnion` │ `FreshTomato` │ `Plate` │ `ChoppedOnion` │ `ChoppedTomato` │ `ChoppedOnion+Plate` │ `ChoppedTomato+Plate` │ `ChoppedOnion+ChoppedTomato` │ `ChoppedOnion+ChoppedTomato+Plate` |
| Behavior b | `Chop(t)` │ `Pick(t)` │ `Merge(t,t)` │ `Serve(t)` │ `WashDirtyPlate()` `PutOutFire()` |
| Conditions c | `h` │ `tautology` |
| Statement s | `While(c):(s)` │ `Parallel:(s_1, s_2, ···)` │ `b` │ `Repeat(s,i)` │ `If(c):(s)` │ `else:(s)` │ |
| Perception h | `is_ordered[t]` │ `is_there[t]` |

## A.2 Multi-Pointer Updater

Algorithm 1 shows the execution rules of parallel programs. The program executor maintains a set of pointers $\mathbb{P}$ pointing to different subroutines. After the agents take the actions, we update each pointer $p$ in $\mathbb{P}$ according to the type of the subroutine that $p$ was pointing to.

---
**Algorithm 1** Multi-Pointer Updater
---

**input** : a set of pointers $\mathbb{P}$, a required repetition number $M$

**for** *each pointer p in* $\mathbb{P}$ **do**

  **if** *p points to an* if-*routine* **then**

    │ Point $p$ to the routine inside/outside the block of primitives in the if-routine if the response to its condition is True/False.

  **end**

  **else if** *p points to an* while-*routine* **then**

    │ Point $p$ to the routine inside/outside the block of primitives in the while-routine if the response to its condition is True/False.

  **end**

  **else if** *p points to a* parallel-*routine* **then**

    │ Split $p$ into multiple pointers, each pointing to a block of subroutines in the parallel-routine.

  **end**

  **else if** *p points to a* repeat-*routine* **then**

    │ Split $p$ into M pointers, each pointing to a copy of the block of subroutines.

  **end**

  **else if** *p points to a behavior primitive and the corresponding subtask is completed* **then**

    **if** *p points to the end of a block of subroutines in a* while-*routine* **then**

      │ Point $p$ to the while-routine.

    **end**

    **else if** *p points to the end of a block of subroutines in a group of blocks of subroutines spawned from a* parallel-*routine/*repeat-*routine*

      **then**

      **if** *the pointed block is not the last remaining one* **then**

        │ remove the pointer

      **end**

      **else**

        │ remove the pointer and generate a new pointer $q$ pointing to the subroutine subsequent to the *parallel*-routine/*repeat*-routine

      **end**

    **end**

    **else**

      │ Point $p$ to the subsequent subroutine. Terminate the program if no subsequent subroutine exists.

    **end**

  **end**

**end**

---

## A.3 Full Illustration of the E-MAPP Algorithm

We describe the whole process of E-MAPP in Algorithm 2, which corresponds to the inference stage of E-MAPP.

---

**Algorithm 2** The E-MAPP Algorithm during test time

---

**input** : Environment $env$ and its guiding program, a set of pointers $\mathbb{P}$ pointing to the possible
        subroutine set, a policy module $f_{policy}$, a perception module $f_{perception}$
**while** *the task is not completed* **do**
    **while** *there are pending perception primitives in the possible subroutine set* **do**
        Run $f_{perception}$ on the perception primitives and get the results $o_{perc}$. Update $P$ according to
        $o_{perc}$ and the rule in Algorithm 1.
    **end**
    **Run** $f_{policy}$ to obtain the auxiliary functions on each behavior primitive $p$ points to.
    **Compute** the cost of each possible allocation based on the auxiliary functions.
    **Run** $f_{policy}$ on the subtasks with minimal cost and obtain the joint action $a$
    **Environment** steps forward with $a$
    **while** *There are completed behavior primitives in the possible subroutine set* **do**
        Update $p$ according to the rule in Algorithm 1
    **end**
**end**

---

## A.4 Training Details for the Policy Module

**Architecture details.** The policy module takes as input a goal vector $g$ and a state $s$ and outputs an action distribution $\tilde{a}$. A goal vector $g$ has a size of 20 where the first 10 elements are one-hot encoded behavior type (e.g., *Chop*) and the latter 10 elements are one-hot encoded behavior arguments (e.g., *Tomato*). A state $s$ is comprised of a map state $s_{\text{map}}$ and an inventory state $s_{\text{inv}}$. The map state $s_{\text{map}}$ has a size of $20 \times 8 \times 8$ where $8 \times 8$ is the resolution of the map and 20 is the number of object types. The inventory state $s_{\text{inv}}$ has a size of 6 where the first two entries denote the location of the agent, the third entry denotes whether the agent is holding objects, and the last three entries denote whether a certain dish is ordered.

The map state $s_{\text{map}}$ is encoded by a four-layer convolutional neural network (CNN) with channel sizes of 32,64,64, and 64. For each convolutional layer, we use 1 as the stride size and 1 as the "same" padding size. Each convolutional layer has a kernel size of 3 except for the first one, which has a kernel size of 5. A ReLU nonlinearity is applied to every convolutional layer. The output is finally flattened into a feature vector and fed into a linear layer, producing a 128-dim map feature vector, denoted as $f_{\text{map}}$.

The inventory state $s_{\text{inv}}$ is encoded by a three-layer MLP with hidden size 128 for all layers. The output feature vector has a dimension of 32, denoted as $f_{\text{inv}}$.

The map feature $f_{\text{map}}$ and the inventory feature $f_{\text{inv}}$ are then concatenated, producing the 160-dim state feature $f_{\text{state}}^1$.

The goal vector $g$ is encoded by a three-layer MLP with a hidden size of 128 for all the layers. The output goal feature $f_{goal}$ is a 640-dim feature vector. $f_{goal}$ can be viewed as the concatenation of four 160-dim feature vectors, denoted as $\gamma_1$, $\beta_1$, $\gamma_2$, and $\gamma_2$ respectively.

The state feature is modulated by $\gamma_1$ and $\beta_1$ as $f_{\text{state}}^2 = f_{\text{state}}^1 \dot{\gamma}_1 + \beta_1$. Next, the modulated feature $f_{\text{state}}^2$ is encoded by a two-layer MLP with both hidden size and output size equal to 128. The output is then modulated by $\gamma_2$ and $\beta_2$, producing the goal-conditioned state feature $f_{\text{state}}$.

Finally, the goal-conditioned state feature $f_{\text{state}}$ is encoded by two linear layers to produce the 24-dim action distribution and the 1-dim value function.

In practice, the parameters of the convolutional layers are shared by the policy net and the value net, while other parameters are separate.

Table 6: **Hyperparameters for the training of the policy module.** We use MAPPO as the backbone algorithm. The common hyperparameters are listed below.

| Name | Value |
|---|---|
| learning rate | 3e-4 |
| training steps | 10M |
| update batch size | 256 |
| number of rollout threads | 8 |
| rollout buffer size | $4096\times 8$ |
| weight of value loss | 0.1 |
| weight of policy loss | 1 |
| weight of entropy loss | 0.01 |

Table 7: **Hyperparameters for the training of the perception module.** We model the perception module as a binary classifier and use cross-entropy loss as the training objective.

| Name | Value |
|---|---|
| learning rate | 3e-4 |
| update batch size | 128 |

In cooperative settings, the goal input of the assistive agent is the leading agent's goal. We use a separate assistive goal encoder for it. The architecture of the altruistic goal encoder is the same as the independent goal encoder mentioned above.

**Self-imitation learning.** In our task domain, it is important to address the challenge of sparse rewards, which is also a key issue for goal-conditioned reinforcement learning [31]. To tackle this, we propose to better utilize the successful trajectories of each agent inspired by self-imitation [32]. For each agent, we select the state-action pairs from the replay buffer with empirical returns that are larger than a threshold $r_{\text{thresh}}$. The self-imitation learning objective of the agent $i$ is $\mathcal{L}_{sil} = -r_i(s, t) \log \pi_i(s_t)$. This loss is added directly to the reinforcement learning loss.

**Hyperparameters.** Table 6 shows the hyperparameters used in the policy module.

### A.5  Architecture and Training Details for the Perception Module.

**Architecture of the perception module.** The architecture of the perception module can be obtained from that of the policy module in Appendix A.4 by replacing the goal encoders with perceptive query encoders. The details of the encoders remain unchanged.

**Dataset collection.** We randomly sample 10k environments as the training dataset. To augment the data, we label the ground truth of the perceptive queries after the agents randomly take 10-15 steps. We leave 10% of the dataset as the evaluation dataset and terminate the training process when the accuracy on the evaluation dataset is larger than 99% five times.

**Hyperparameters.** Table 7 shows the hyperparameters used in training the perception module.

### A.6  Training Details for the Auxiliary Functions

We randomly sample 10k environments as the training environments for each of the three auxiliary functions: $f_{\text{reach}}$, $f_{\text{feas}}$, and $f_{cost\text{-}to\text{-}go}$.

To train the reachability function, we run the pre-trained single-agent policy on the training environments to collect the training data. The agent is required to fulfill a specific subtask within an episode of 128 timesteps. If the agent violates the program by completing the wrong subtask or exceeding the time limit, we will label every state in the trajectory as $False$. In contrast, if the agent successfully completes the subtask, we will label the states as $True$. We train the reachability function by alternatively collecting (state, goal, label) triplets and training on the collected data.

To train the feasibility function, we run the pre-trained multi-agent cooperative policy in the training environments to collect the training data. The agents are required to complete a specific subtask within an episode of 128 timesteps. Although both agents are responsible for the subtask, we only take account of the leading agent's trajectories. The states in the trajectories are labeled True/False according to whether the subtask is successfully completed during an episode.

To alleviate the problem of mistakenly labeling a feasible (state, goal) pair as *False* due to the imperfection of the pre-trained policy, we run multiple times on the environment that is labeled *False* and correct the label if there is a successful case.

## A.7 Detailed Environment Description

In the *Overcooked* environment, the goal of the agents is to complete long-horizon tasks, such as preparing a dish. A typical dish would require first picking up ingredients from certain supplies, then processing the ingredients (e.g., chop, merge together, or put on a plate), and finally delivering the dish (and washing the plates if another dish is still needed). We also introduce "on-fire" as an accident for the agents to handle with a fire extinguisher. The subtasks correspond to the subroutines in the domain-specific language (DSL). Different subroutines, together with the control flow, form the guiding parallel programs for the agents.

For each agent, the state is composed of a map state and an inventory state. Both the maps and the programs are one-hot encoded as object-centric representations. The agents' action space is discrete. There are 24 possible actions, including 6 operations (move, pick, place, serve, merge, and interact) in 4 directions. In multi-agent settings, we follow the original game and assume full observability.

## A.8 More Visualizations for Generalization

Figure 7 shows more detailed visualizations of two emergent behaviors.

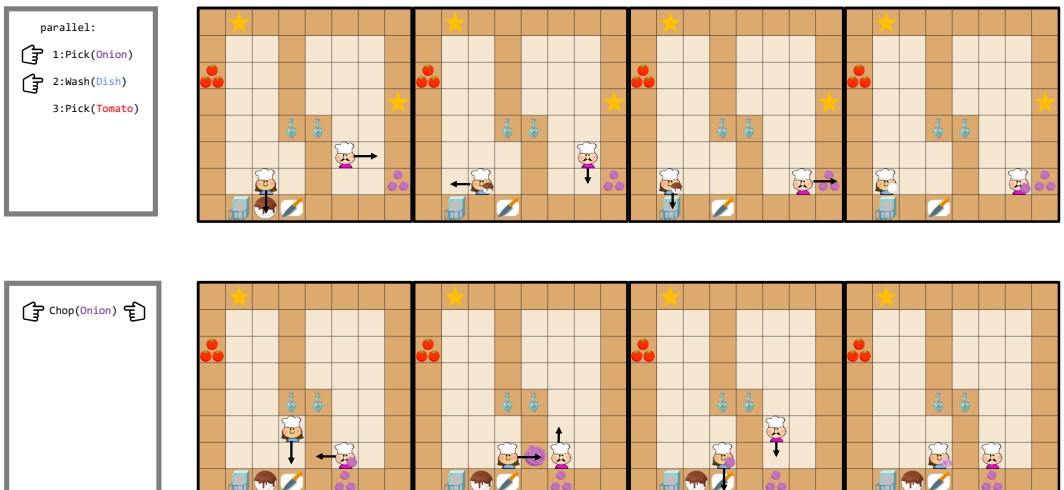

Figure 7: **Visualizations of parallel executions and cooperative behaviors.** In the first example, there are three feasible subtasks. The blue agent chooses the subtask *Wash(Dish)* because the dirty plate is both reachable from and close to him. The magenta agent chooses the subtask *Pick(Onion)* because the onion supply is reachable from him. In the second example, the two agents cooperate on the subtask *Chop(Onion)*. The magenta agent picks the onion and passes it to the blue agent, while the blue agent finalizes the subtask by chopping the onion at the chopping block.

## A.9 Baseline Description

In our study, we mainly consider two types of baselines: state-of-the-art MARL algorithms and natural language-guided agents.

- State-of-the-art multi-agent reinforcement learning. We directly use MAPPO [51] to train joint policies without the guiding program, while still providing dense rewards to the agents when any correct subtask is completed;

- Natural language-guided agent. We train a goal-conditioned policy where the goal is encoded from the natural language description with a pre-trained BERT model in the PyTorch package *transformer* [10]. The encoded features of the tokens are average-pooled and frozen during the training process. Then we use a learnable MLP to encode the frozen features into goal features. The MLP has three layers connected by the ReLU activation, and the hidden sizes of the MLP are all 128. We note that here we do not provide a program-like structure for the language-guided agents.

The baseline models have the same policy and value network architecture as those of E-MAPP. When training the end-to-end baseline models, the agents are rewarded $0.2$ if they complete a correct subtask and $1$ if the final task is completed. We also use self-imitation learning in baseline algorithms to address the sparse reward problem.

The computation costs of E-MAPP and baselines are shown in Table 8

Table 8: **Computation Cost.** The number of parameters and the running time in E-MAPP and baselines.

| Model | Parameters | Running Time |
|---|---|---|
| E-MAPP | 5.6M | around 72h |
| natural language guided agent | 2.3M | around 36h |
| MAPPO | 1.8M | around 24h |

## A.10 Results in the Stacking Environment

To evaluate the ability of E-MAPP to scale to more complex control tasks, we also demonstrate how E-MAPP works on a parallel stacking task set. It is desired that the two franka arms cooperate to complete two stacks of blocks in a given order. The behavior primitives considered in this setting are in the form of *Stack(c, idx)*, which represents the subtask of putting a $c$-colored block on the top of the $idx$-th pile. We use motion planning as the policy to complete the subtasks. Figure 8 is a demonstration of the task completion process. The baselines considered above fail to achieve the goal in any episode, while E-MAPP can achieve a 46% completion rate.

## A.11 Computational Resources

We train our model on a single Nvidia TITAN-X GPU, in a 16-core Ubuntu 18.04 Linux server.

## A.12 Examples of Programs Used in Evaluation

### A.12.1 Easy tasks

```
if IsOnFire():
    PutOutFire()

if is_ordered(ChoppedTomato):
    Serve(ChoppledTomato+Plate)

if is_ordered(ChoppedOnion):
    Pick(FreshOnion)
```

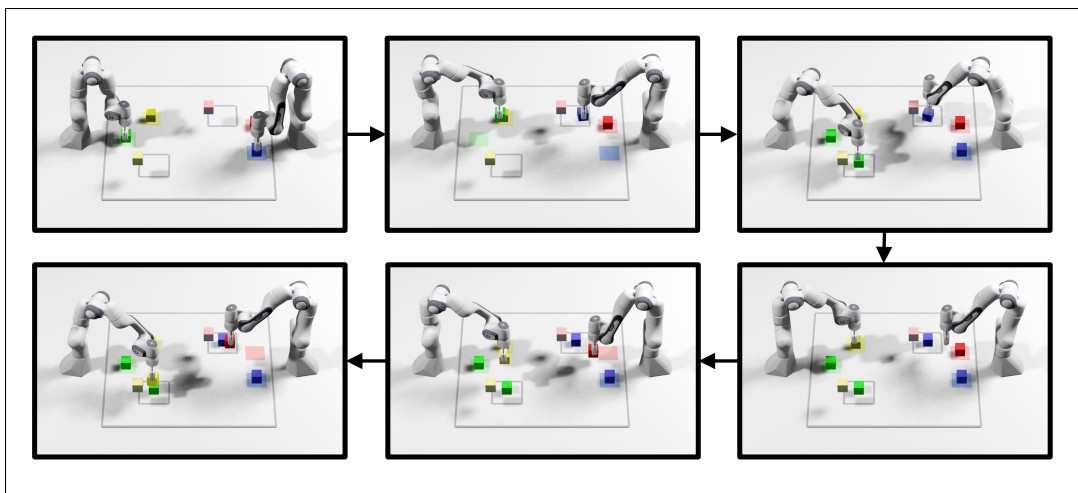

Figure 8: **A demonstration of the completion process of stacking tasks.** The pictures show the process of picking up boxes and stacking them.

### A.12.2 Medium tasks

```
repeat:
    Pick(FreshTomato)

parallel:
    1. Pick(FreshOnion)
    2. Pick(FreshTomato)
    3. WashDirtyPlate()

if is_ordered(ChoppedTomato):
    Chop(FreshTomato)
    Merge(ChoppedTomato,Plate)
    Serve(ChoppedTomato+Plate)
```

### A.12.3 Hard tasks

```
parallel:
    1: if is_ordered(Onion):
            Merge(ChoppedOnion,Plate)
            Serve(ChoppedOnion)
    2: if is_ordered(Onion):
            Merge(ChoppedTomato,Plate)
            Serve(ChoppedTomato)
    3: while (True):
        If (IsOnFire()):
            PutOffFire()

parallel:
        1:
            Pick(FreshOnion)
            Chop(FreshOnion)
        2:
            Pick(FreshTomato)
            Chop(FreshTomato)
        3:
            WashDirtyPlate()
        4:
            Merge(ChoppedOnion,Plate)
```

```
              Serve(ChoppedOnion)
    5:
              Merge(ChoppedTomato,Plate)
              Serve(ChoppedTomato)
```

## A.13   More experiments

### A.13.1   Overcooked with more agents

We conduct an experiment with a doubled number of agents to evaluate our algorithm. Table 9 shows the results. The results indicate that E-MAPP can scale to environments with more agents and further boost the time efficiency by parallelization.

Table 9: **Additional experiment in Overcooked involving four agents.** E-MAPP can scale to environments with more agents and further boost the time efficiency by parallelization.

| model | score | completion rate |
|---|---|---|
| E-MAPP(original) | 0.99±0.22 | 43.7% |
| E-Mapp(larger) | 1.13± 0.31 | 46.3% |

### A.13.2   Comparison with other centralized execution agents

We also compare E-MAPP with a centralized execution approach. We implement a centralized PPO where joint policy is directly produced by a centralized network. Table 10 shows the results. The results indicate that the centralized PPO suffers from the high dimensionality of the joint action space and fails to learn cooperation and coordination.

Table 10: **Comparison of E-MAPP, centralized algorithm PPO and decentralized algorithm MAPPO.** We additionally compare E-MAPP with a centralized PPO that directly outputs the joint policy.

| model | score |
|---|---|
| E-MAPP | 1.58±0.60 |
| MAPPO(decentralized) | 0.59± 0.27 |
| MAPPO(centralized) | 0.48 ± 0.21 |

## A.14   Potential parallel-program synthesis approaches

The guiding program in our work can be obtained with program synthesis approaches. When it comes to a new domain, we can devise new perception primitives and behavior primitives based on object properties and interactions among objects [25]. These primitives, along with the branching and parallelization keywords, compose the DSL. Previous approaches on program synthesis [44, 11, 7, 8] can be applied to synthesize programs for tasks. For example, we can synthesize programs from diverse video demonstrations. The activities (subtasks) of a task in a video can be segmented out as a subroutine for program extraction [44]. By summarizing the chronological order of subtask completions, we can obtain the dependence of subtasks and put the possibly parallelizable subtask in one parallel subroutine.