# OpenReview forum: "E-MAPP: Efficient Multi-Agent Reinforcement Learning with Parallel Program Guidance"
_NeurIPS.cc/2022/Conference — NeurIPS 2022 Accept_

### Official Review · Reviewer_GMQT · 2022-07-08

**Rating:** 6
**Confidence:** 3
**Soundness:** 3 good
**Presentation:** 3 good
**Contribution:** 3 good

**Summary:**

This work introduces Efficient Multi-Agent Reinforcement Learning with Parallel Program Guidance (E-MAPP), a methodology designed to enable multi-agent, long-horizon task completion. Toward this goal, E-MAPP structures tasks as (parallelizable) programs using a hand-designed domain-specific language (DLS) which specifies behavior subtasks (e.g. chop a tomato) and "perception primitives" (e.g. is anything on fire); a top-level controller then uses these perceptual primitives, along with the given program, to assign agents to subtasks so as to efficiently complete the overall task. E-MAPP is evaluated using a novel extension of the Overcooked gridworld from prior work (in Overcooked, multiple agents must work together to compose dishes. Compared to competing baselines (MAPPO and a variant of MAPPO with language guidance), E-MAPP produces agents that are substantially more competent at solving challenging, long-horizon, tasks in the Overcooked environment even when those tasks were not seen during training.


**Questions:**

Currently, I lean towards rejection as the weaknesses I've listed above seem quite substantial when considering applying E-MAPP to a new domain. I would be more than happy to increase my rating if strong arguments could be given as to why I might be mistaken. I would be particularly interested in hearing responses to my concerns regarding the number of pretrained components and the requirements for full observability / low dimensional states. I have also included a few, more minor, questions in my line-by-line notes above.


**Limitations:**

Assuming the weaknesses I've flagged above are true weaknesses, it would be great to be upfront and add these to the limitations section. I don't believe there are any substantive ethical concerns for this work.


**Strengths And Weaknesses:**

## Strengths

This paper studies an interesting and challenging problem (long-horizon planning in the cooperative multi-agent setting). The proposed E-MAPP framework makes several non-trivial extensions to the idea of program-guided single-agent task completion introduced in prior work. The presented empirical results suggest that E-MAPP can be quite effective, especially when compared to standard "end-to-end" multi-agent RL methods (e.g. MAPPO) which do not use programs as an inductive bias. The paper is well-written grammatically although, as I note below, some work can be done to improve overall clarity.

## Weaknesses

While I very much appreciate the core goal of this paper, I do believe there are a few weaknesses that temper my enthusiasm (described below).

- Many pretrained components

This work appears to depend on a very large number of pretrained components: perception modules, single-agent policies, and each of the three auxiliary functions (reachability, feasibility, and cost-to-go). That these all must be trained in different phases with separate datasets and different training techniques (e.g. self-imitation learning) seems to make it quite challenging to E-MAPP in new settings.

- Efficiency of subtask allocation

The current strategy for subtask allocation, from my understanding, requires searching over a search space that grows exponentially in the number of tasks and agents (M^N possible allocations of N agents to M tasks). This appears to limit the applicability of E-MAPP to tasks with larger numbers of agents.

- Fully observed and low-dimensional

The need to train many separate components to high levels of accuracy suggests that E-MAPP may not work well in complex, high-dimensional, environments (e.g. learning from pixels or learning in partially observed environments) where even training the perception modules may be very challenging.

- Clarity

While the paper is quite easy to read from a grammatical standpoint, I found several parts of the paper to be needlessly vague. For instance:

1.  The continuous control experiment is essentially completely unexplained, no results are even mentioned in the main paper and the appendix provides little additional detail.
2. The inputs given to the various baseline models (MAPPO and the model with NL guidance) is unclear in the main paper. Even reading the appendix I'm unsure how the goal is encoded and given to the MAPPO agent.
3. Several figures take up quite a lot of space (e.g. Fig 2/3/5) while the heart of the paper, namely the description of how exactly the E-MAPP framework works (e.g. pointer allocation and when/how the auxiliary functions are used) is largely relegated to the appendix.

## Line-by-line notes

Here are a few minor line-by-line notes:

- Line 33
    * "different feasibility" sounds odd.

- Line 64
    * Factorization approaches have also been investigated on the policy side, see e.g. "A Cordial Sync: Going Beyond Marginal ..." Jain et al. (ECCV'20)

- Line 65
    * It seems a bit strange to not highlight imitation learning, and related ideas, as an approach for solving long-horizon tasks and the sparse reward problem. E.g. generative adversarial imitation learning.

- Line 152-153
    * "...to a subtask that requires to be..." -> "...to a subtask that must be..."

- Line 155
    * "a perceptive query is responded" -> "a response to a perceptive query is received"

- Lines 156-159
    * Things are quite vague, give an example.

- Lines 181-182
    * An ablation of this choice would be nice.

- Line 208
    * "reducing the binary" -> "minimizing the binary"

- Lines 227-228
    * Why apply log to feasibility and reachability and not cost-to-go?

- Lines 247-249
    * The phrasing here is a bit confusing, it almost seems like your extension is a whole new environment. You should rephrase to make more clear that you didn't build a new gridworld but you adapted an existing one.

- Line 259
    * What's the definition of "novel" here?

- Line 266
    * "accumulative" -> "cumulative"

- Line 292
    * "find removing" -> "find that removing"

- Table 3
    * The error bars for the average scores seem huge?

- Appendix Lines 562-567
    * Do you also use self-imitation learning with the MAPPO baseline? If not, what is the impact of self-imitation learning alone?

- Appendix Lines 624
 - "sclae" -> "scale"

---

> ### Author Response · Authors · 2022-08-02
> **Response to Reviewer GMQT(part 1/2)**
>
> Dear reviewer GMQT, thank you for your detailed and thorough review.  We seek to address each of your concerns with the following responses:
>
> Q: This work appears to depend on a very large number of pretrained components: perception  modules, single-agent policies, and each of the three auxiliary functions (reachability, feasibility, and cost-to-go). That these all  must be trained in different phases with separate datasets and different training techniques (e.g. self-imitation learning) seems to make it  quite challenging to E-MAPP in new settings.
>
> A: Although our framework is modular, the training process is natural. The perception module and the auxiliary functions are all trained in a supervised manner, which is efficient (within several hours) and accurate (with validation accuracy over 90%). The training techniques such as self-imitation used in E-MAPP policy learning are also commonly used in standard multi-agent RL algorithms including MAPPO.
>
> Q: The current strategy for subtask allocation, from my understanding, requires searching over a search space that grows exponentially in the number of tasks and agents (M^N possible allocations of N agents to M tasks). This appears to limit the applicability of E-MAPP to tasks with larger numbers of  agents.
>
> A: With some assumptions, we can obtain a polynomial-time subtask allocator that scales well.  Please refer to the "Computational complexity of subtask allocation" section in the general response for further details.
>
> Q: Fully observed?
>
> A: Although the experiments are conducted in a fully observable setting, E-MAPP can also be applied to partially observable environments. Please refer to the "Why fully observability" section in the general response.
>
> Q: The need to train many separate components to high levels of accuracy suggests that E-MAPP may not work well in complex, high-dimensional, environments (e.g., learning from pixels or learning in partially observed environments), where even training the perception modules may be very challenging.
>
> A: While E-MAPP does require a few pretrained components, we argue that it can scale well even in the face of complicated tasks. Specifically
>
> 1) For branching subroutines (e.g., IsOnFire()), we keep a memory to track and constantly check whether the previous branching output is correct. For example, if the perception module erroneously outputs True for IsOnFire() under some unseen states, in the next few steps, it will correct its own error with high probability and guide the agent to the correct branch promptly.
> 2) We use a hyperparameter “maximum timesteps of tries” in the subtask allocator to prevent agents from attempting unfeasible subtasks. This reduces the impact of inaccurate feasibility/reachability functions.
>
> Q: Explain the continuous control experiment.
>
> A: The continuous control task requires two arms to concurrently stack two piles of blocks.  We include this task for showing that parallel programs can also be extended to continuous control, and are useful tools in other domains. A program example is
> ```
> parallel:
>     stack(yellow, green) # stack yellow block on green block
>     stack(red, blue)
> ```
> Q: The inputs given to the various baseline models (MAPPO and the model with NL guidance)?
>
> A: The MAPPO agent is trained separately for each task. The natural language-guided model takes in a goal described in natural language and an observation. Then, it learns a goal-conditioned policy based on MAPPO. The goal is encoded with a pretrained sequence encoder(BERT) and a learnable MLP that shares structure with that in E-MAPP. We will add a clearer description of the NL model in the paper.
>
> Q: Several figures take up quite a lot of space (e.g. Fig 2/3/5) while the heart of the paper,  namely the description of how exactly the E-MAPP framework works (e.g.  pointer allocation and when/how the auxiliary functions are used) is largely relegated to the appendix.
>
> A: We have adjusted the layouts of the paper in the revised version.
>
> Q: Line 33 "different feasibility" sounds odd.
>
> A: We have replaced it with “different abilities”.
>
> Q: Line 64 Factorization approaches have also been investigated on the policy side, see e.g. "A Cordial Sync: Going Beyond Marginal ..." Jain et al. (ECCV'20)
>
> A: Thanks for your suggestion for the reference. We have cited and discussed these papers in the revised version.
>
> Q: Highlight imitation learning, and related ideas, as an approach for solving long-horizon tasks and the sparse reward problem. E.g. generative adversarial imitation learning.
>
> A: Thanks for your suggestion for the reference. We have cited and discussed these papers in the revised version.

---

> ### Author Response · Authors · 2022-08-02
> **Response to Reviewer GMQT(part 2/2)**
>
> Q: Lines 156-159 Things are quite vague, give an example.
> A: Here is an example explaining the program executor’s rule. If the program pointer points to “Serve(Onion)”, but a dish of tomato is served(wrong subtask is completed!), the program executor will terminate the program and end the current episode.
> ```
> If (is_ordered(Onion)):
>     Serve(Onion)  < - - - - - - - - - - - - - - -
> Else:
>     Serve(Tomato)
> ```
>
> Q: Lines 181-182 An ablation of this choice would be nice. (Do you also use self-imitation learning with the MAPPO baseline? If not, what is the impact of self-imitation learning alone?)
>
> A: Yes. We use self-imitation in the MAPPO agent and the NL-guided agent. We will add it to the paper.
>
> Q: Why apply log to feasibility and reachability and not cost-to-go?
>
> A: Intuitively, an unfeasible subtask should not be allocated to any agent. The feasibility and reachability ranges from 0 to 1. Applying log to the feasibility function and the reachability function can bring a huge cost to an unfeasible subtask, thus preventing it from allocating to any agent.
>
> Q: What's the definition of "novel" here?
>
> A: The maps in the testing environments are unseen during training, thus requiring the agent to generalize to new maps.
>
> Q: Table 3 The error bars for the average scores seem huge?
>
> A: Yes. The testing maps are diversified(i.e., object positions are random), leading to a huge variance among scores on different maps.
>
> Misc: We have fixed the typos and the improper phrasing you mentioned.

---

> ### Author Response · Authors · 2022-08-08
> **Follow-up response to Reviewer GMQT**
>
> We would like to first thank you again for your constructive comments and helpful suggestions. Since we are near the end of the discussion phase, we would like to post a follow-up response.
> In our previous response and our revision, we have addressed your following concerns:
>
> - We have conducted additional experiments and added additional analysis to show that our method can scale to more complex environments. (e.g., partially-observable environments, environments with larger number of agents)
> - We further refined the paper by fixing the typos and clarifying some facts.
>
> Do you find our responses satisfactory? Or if you have any other suggestions on further improving the manuscript, it would be great if you can post them during the discussion phase. We are more than happy to add them to our paper and submit a new revision before the discussion phase ends. Thank you!

---

> > ### Comment · Reviewer_GMQT · 2022-08-08
> > **Re: Follow-up response to Reviewer GMQT**
> >
> > Thank you for your extensive rebuttal and additional experimentations; I especially appreciate the results showing performance in the partially-observed setting. As several of my proposed weaknesses have been addressed I will increase my score to a 6. I struggle to go above this score for many of the same reasons addressed by Reviewer KHSD: it seems like applying E-MAPP to a new setting will require a huge amount of work (e.g. defining primitives, pretraining modules, and determining which assumptions can be leveraged to lower subtask allocation time).

---

### Official Review · Reviewer_KHSD · 2022-07-09

**Rating:** 6
**Confidence:** 3
**Soundness:** 1 poor
**Presentation:** 3 good
**Contribution:** 2 fair

**Summary:**

The paper introduces a new method “E-MAPP” , a framework which utilizes a centralized program to issue subtasks to agents within a team to achieve tasks which require long-term planning. EMAPP contains a central programmatic controller which issues goals to a team of subagents. These agents are goal-conditioned policies who are trained specifically to solve these subgoals. The paper evaluates their claims on the Overcooked environment and evaluates against baselines such as MAPPO. Another notable addition is the increased complexity added to the game dynamics within the Overcooked environment.


**Questions:**


Mostly stated above the in the weaknesses section:

1. I find it very hard to understand what the Program Executor actually suggests or how this is synthesised. Can you provide examples of the programs devised.

2. Analysis of queries asked and performance of the state encoder on producing correct answers would be useful to understand how well the Program Executor /State Encoder is working.

3. Understanding the number of unfeasible tasks produced would be useful.

4. It is unclear why E-MAPP performs imperfectly in the easy environment. Can the authors add some comments on this.

5. Ablation analysis of E-MAPP (as it has many components) in general would be appreciated, it's unclear where to attribute performance gains to currently.



**Limitations:**

It would be useful to discuss limitations in computation resources and scalability of this approach. Overcooked is neither a game of many players or many coordination conventions.


**Strengths And Weaknesses:**

The paper is well written and the diagrams are exceptionally useful to articulate the approach being taken. The approach seems original (in the context of MultiAgent systems) however I can not comment on how novelly it extends Program use within Single Agent RL.

My main criticism of this work is the sheer complexity of the proposed approach, the limited analysis of its improvement over baselines, and its overall usefulness.

1. It is unclear how much information is arbitrarily being hardcoded into the program executor (what decides the perception and behavior primitives), nor is it clear how you devise the Possible Subroutine Set. Hardcoding such components reduces the ability for this approach to be applied to any novel task. Thus this approach does not help discover/explore novel solutions to the game of Overcooked.

2. To further this, the additional auxiliary tasks used to shape the policy network (feasibility, reachability, cost-to-go) are also hardcoded heuristics to evaluate sub-agent fitness for a task. In new problem domains it is unclear how you can devise this. In particular the Criteria for subtask allocation confuses me. Surely simple reward maximizing would lead to this criteria naturally arising within the Task Allocator.

3. There is no discussion on the computationally complexity or feasibility of this approach versus baselines. In comparison to the baselines this seems to contain significantly more networks, a search for legal subtask allocation which scales O(n!) to the number of subtasks and finally a curriculum which requires pre-training a state on ground truth sub-goals, training n agents separately and finally then training the joint n-agents.

4. Given the complexity of E-MAPP it is unclear where to attribute performance gains to. In particular the baseline of “Natural language–guided model” is never explained. A comparison of the goals provided by the Natural Language (or vocab size) vs the Task Allocator (and number of primitives) would be useful.

5. E-MAPP fundamentally is a centralised execution algorithm (a joint policy is learnt) - so to compare to decentralised execution algorithms such as MAPPO seems disingenuous. A much better comparison would be a centralised agent. Such a comparison would help justify the claim of “time efficiency” presented in the abstract.

6. Given the sheer amount of biases and knowledge bestowed into E-MAPP it seems trivial to outperform MAPPO or other methods on discovering winning strategies for harder tasks.

7. The choice of rewards (0.2 for subtask, 1 for task) provided in the “Average Scores” seem arbitrary. Please provide reasoning for these.

---

> ### Author Response · Authors · 2022-08-02
> **Response to Reviewer KHSD(part 1/3)**
>
> Dear reviewer KHSD, thank you for your detailed and thorough review.  We seek to address each of your concerns with the following responses:
>
> Q: The approach seems original (in the context of MultiAgent systems) however I can not  comment on how novelly it extends Program use within Single Agent RL.
>
> A: E-MAPP is more than a simple extension of program use with single agent RL. We propose original approaches to tackle the following unique challenges: 1) Subtask structure discovery. We combine parallelization keywords and the feasibility function to automatically infer the parallelizable subtasks, while single agent RL can only strictly follow the order of program subroutines. 2) Allocating subtasks to agents. We design three auxiliary functions as the criteria for subtask allocation. However, in program-guided single agent RL settings, the sequential program provides only one subtask each time for the agent, so there is no need to match different subtasks to different agents. 3) Cooperative policies. We design a lead-assist framework for cooperative subtasks to address the resource racing problem, which is also a non-existent problem in single agent RL settings.
>
> Q: Information hardcoded into the program executor (what decides the perception and behavior primitives), and the design of the Possible Subroutine Set.
>
> A: We have listed the DSL used in the overcooked environment in Appendix A.1.  The list involves all the domain-specific information. The elements of the possible subroutine set are primitives listed in the DSL. There is no additional information encoded other than what is mentioned above.
>
> Q: Hardcoding such components reduces the ability for this approach to be applied to any novel task. Thus this approach does not help discover/explore novel solutions to the game of Overcooked.
>
> A: The DSL describes only the basic operations and necessary procedures in the game. The low-level policies and the subtask planning are all learned. Discovering novel solutions such as good cooperative policies(e.g., one agent help another agent by delivering tools) and time-efficient task planners(e.g., raising priority on subtasks that are preconditions of other subtasks) will not be hindered.
>
> Q: To further this, the additional auxiliary tasks used to shape the policy network  (feasibility, reachability, cost-to-go) are also hardcoded heuristics to evaluate sub-agent fitness for a task. In new problem domains it is unclear how you can devise this.
>
> A: We respectfully disagree that the auxiliary functions are specially devised heuristics for the game Overcooked. The feasibility function filters out the subtasks with uncompleted predecessor subtasks, the reachability function denotes whether a subtask is cooperative or non-cooperative. The cost-to-go function can be trained to fit any cost functions in the new domain. These functions and the training process are general for different tasks, and can all be transferred to new domains without extra effort. For example, in the multi-stacking environment mentioned in Appendix A.10, the reachability function refers to whether the target block is within the arm’s reach, while the feasibility function refers to whether the precondition (e.g., B is in the right position) is satisfied for a subtask (e.g., stack A on B).

---

> > ### Comment · Reviewer_KHSD · 2022-08-07
> > **Update score**
> >
> > Thank you for your extensive rebuttal!
> >
> > My concerns have been sufficiently addressed to warrant bumping this up to a 6.
> >
> > I still think this approach is limited by knowing the correct subtask primitives, and look forward to further work addressing this.

---

> ### Author Response · Authors · 2022-08-02
> **Response to Reviewer KHSD(part 2/3)**
>
> Q: The criteria for subtask allocation. Surely simple reward maximizing would lead to this criteria naturally arising within the Task Allocator.
>
> A: Our approach aims at discovering solutions for generalizable and composable tasks instead of a single task. The proposed criteria for subtask allocation rules out the unfeasible subtasks and greedily choose the subtask with smallest cost conditioned on current states. It is a general criteria for tasks with any subtask structure. In contrast, an end-to-end reward maximization approach may fail to discover general policies for different tasks. This is supported by the experiment on NL agents, in which the complete task is embedded as a goal and the agents are required to maximize the cumulative reward in a complete episode. The results show that end-to-end reward maximizing may fail to learn efficient task-planners for multi-tasks.
>
> Q: Computationally complexity or feasibility of this approach versus baselines. In comparison to the baselines this seems to contain significantly more networks, a search for legal subtask  allocation which scales O(n!) to the number of subtasks and finally a curriculum which requires pre-training a state on ground truth sub-goals, training n agents separately and finally then training the joint n-agents.
>
> A: With some assumptions, we can obtain a polynomial-time subtask allocator that scales well. Please refer to the "Computational complexity of subtask allocation" section in the general response.
>
> Q: Explain the baseline of “Natural language–guided model”. A comparison of the goals provided by the Natural Language  (or vocab size) vs the Task Allocator (and number of primitives) would be useful.
>
> A: The natural language-guided model takes in a goal described in natural language and an observation. Then, it learns a goal-conditioned policy based on MAPPO. The goal is encoded with a pretrained sequence encoder (BERT) and a learnable MLP that shares structure with that in E-MAPP. We havel added a clearer description of the NL model in the paper in Appendix A.9. The vocab has a size of 30, containing task description words and conjunctive words (e.g., if, then, while). The number of possible primitives is 24, including compositions of behavior types and subjects. Here is an example
> |       task in program       |      task in natural language       |
> | :-------------------------: | :---------------------------------: |
> | if IsOnFire():  PutOutFire() | If there is fire, put out the fire. |
>
> Q: E-MAPP fundamentally is a centralized execution algorithm (a joint policy is learnt) - so … a much better comparison would be a centralized agent.  Such a comparison would help justify the claim of “time efficiency” presented in the abstract.
>
> A: The MAPPO agents share the same observation in our setting, so cooperative behaviors are likely to emerge. We add another experiment to compare E-MAPP with centralized agents. We implement a centralized PPO where joint policy is directly produced by a centralized network. The results indicate that the centralized PPO suffers from the high-dimensionality of the joint action space and fails to learn the cooperation and the coordination.
> |        model         |    score    |
> | :------------------: | :---------: |
> |        E-MAPP        |  1.58±0.60  |
> | MAPPO(decentralized) | 0.59± 0.27  |
> |  MAPPO(centralized)  | 0.48 ± 0.21 |
>
>
> Q: Significance of E-MAPP’s performance vs. MAPPO or other methods on discovering winning strategies for harder tasks.
>
> A: E-MAPP improves two types of fundamental challenges in multi-agent settings: solving long-horizon tasks efficiently and performing compositional generalization. In trade off, we devise a domain-specific language to describe the structures of the tasks. The DSL is very natural to design from task knowledge. For example, we only hint the agents with a $\textit{parallel}$ keywords from DSL, while the agents learn to find an efficient way to execute the programs automatically.  Hence, considering the performance gain, we argue that the introduced inductive biases are reasonable and worth the merits.
>
> We believe this is a promising direction to learn more inductive biases from videos or narrations. Our work opens this welcoming avenue in the multi-agent setting.
>
> Q: The choice of rewards (0.2 for subtask, 1 for task) provided in the “Average Scores” seem  arbitrary. Please provide reasoning for these.
>
> A: We empirically use a smaller value for dense reward and a larger value for the final goal.  We note that the baselines (MAPPO and NL-guided MAPPO) also used the reward 0.2 for subtask completion and 1 for the whole task completion.

---

> ### Author Response · Authors · 2022-08-02
> **Response to Reviewer KHSD(part 3/3)**
>
> Q: What the Program Executor actually suggests or how this is synthesized? Can you provide examples of the programs devised?
>
> A: We have added another section "Examples of Programs Used in Evaluation" in Appendix A.11 to show more program examples. An example is as follows:
> ```
> parallel:
>     1: if is_ordered(“Onion”):
>             Merge(“ChoppedOnion”,”Plate”)
>            Serve(“ChoppedOnion”)
>     2: if is_ordered(“Onion”):
>             Merge(“ChoppedTomato”,”Plate”)
>            Serve(“ChoppedTomato”)
>     3: while (True):
>         If (IsOnFire()):
>             PutOffFire()
> ```
>
> Q: Analysis of queries asked and performance of the state encoder on producing correct answers would be useful to understand how well the Program Executor /State Encoder is working.
>
> A: The accuracy of correct answers to the queries are about 99.2%. We will add this to the paper.
>
> Q: Understanding the number of unfeasible tasks produced would be useful.
>
> A: The subtask allocator tasks input 3-5 subtasks, and about 50% of them are unfeasible on average.
>
> Q: Why E-MAPP performs imperfectly in the easy environment? Can the authors add some comments on this.
>
> A: The MAPPO algorithm is not a goal-conditioned RL algorithm; we train it from scratch on each of the tasks. This training procedure gives MAPPO the advantage to overfit in short-horizon tasks while E-MAPP uses a single goal-conditioned policy for all the tasks. However, when the complexity of the tasks grows large, MAPPO  fails to learn meaningful behaviors even with the advantage.
>
> Q: Ablation analysis of E-MAPP (as it has many components) in general would be appreciated, it's unclear where to attribute performance gains to currently.
>
> A: We have conducted ablation experiments on the key components in Section 5.3, showing that both the parallel program executor and the auxiliary functions (feasibility, reachability, cost-to-go) for subtask allocation are important. The perception module and the policy module are commonly used modules in the program-guided-agent framework, which are also indispensable components.
>
> Q: It would be useful to discuss limitations in computation resources and scalability of this approach. Overcooked is neither a game of many players or many coordination conventions.
>
> A: With some assumptions, we can obtain a polynomial-time subtask allocator that scales well to the number of agents and subtasks. E-MAPP can also be adapted to new domains. Please refer to section"Computational complexity of subtask allocation" and section "How to scale to new domain" in the general response for further details.

---

### Official Review · Reviewer_uuPM · 2022-07-09

**Rating:** 6
**Confidence:** 4
**Soundness:** 3 good
**Presentation:** 3 good
**Contribution:** 3 good

**Summary:**

This paper addresses the problem of learning to fulfill a task described by programs designed for parallelization with multiple agents. To this end, the paper proposes a framework that can infer the structure of parallelism from programs and efficiently allocate subtasks by enforcing cooperation and division of labor among agents. The experiments on the overcooked domain, where agents need to collaborate to make dishes, show that the proposed framework outperforms baselines and achieves higher task completion rates and better generalization. Ablation studies suggest that the proposed feasibility predictor, reachability predictor, and cost-to-go predictor all contribute to the improved performance. I believe this work studies a promising problem and presents an interesting framework with sufficient evaluation. Yet, I still have some concerns detailed in the following section.

**Questions:**

Described in Strengths And Weaknesses section.

**Limitations:**

Described in Strengths And Weaknesses section.

**Strengths And Weaknesses:**

## Paper strengths and contributions
**Motivation and intuition**
I believe learning to fulfill a task described by programs designed for parallelization with multiple agents is an important problem and has a wide range of applications.

**Novelty**
To the best of my knowledge, this is the first work researching following program instructions with multiple agents.

**Technical contribution**
- The DSL used in this work seems like a reasonably good DSL for describing the overcooked domains.
- The proposed task allocator learning reachability, feasibility, and cost-to-go seems effective.

**Clarity**
The overall writing is clear.

**Ablation study**
Ablation studies are helpful for understanding thecontributionsn of each component (i.e. reachability, feasibility, and cost-to-go) of learning to allocate tasks.

**Experimental results**
The experimental results show that the proposed framework outperforms a state-of-the-art multi-agent reinforcement learning baseline (MAPPO) and a baseline that is instructed by natural language description.

## Paper weaknesses and questions

**Fully observability**
While not explicitly stated in the paper, this work seems to assume the agents operate in a fully observable environment and the same observation is shared across all agents. This could fundamentally limit applying the proposed framework to more realistic domains.

**Task example**
There are very few task (program) examples given in the main paper and the supplementary material. It is tough to judge the performance of the proposed framework and the baselines without knowing the tasks used to evaluate them. How easy are the easy tasks? How hard are the hard tasks? How different are the tasks used for learning and the tasks used for evaluating zero-shot generalization?

**How to obtain tasks/programs**
This work studies how to fulfill tasks described by programs. It would make more sense to shed some light on how such programs can be obtained in the first place to motivate the problem. I suggest the authors include a discussion on program synthesis works that aim to produce such programs, such as RobustFill: Neural Program Learning under Noisy I/O (ICML 2017), Neural Program Synthesis from Diverse Demonstration Videos (ICML 2018), Execution-Guided Neural Program Synthesis (ICLR 2019), Learning to Describe Scenes with Programs (ICLR 2019), Latent Execution for Neural Program Synthesis (NeurIPS 2021), etc.

---

> ### Author Response · Authors · 2022-08-02
> **Response to Reviewer uuPM**
>
> Dear reviewer uuPM, thank you for your detailed and thorough review.  We seek to address each of your concerns with the following responses:
>
> Q: While not explicitly stated in the paper, this work seems to assume the agents operate in a  fully observable environment and the same observation is shared across all agents. This could fundamentally limit applying the proposed framework to more realistic domains.
>
> A: E-MAPP can also be applied to partially observable environments. Please refer to the "why fully observability" section in general response for further details.
>
> Q: There are very few  task (program) examples given in the main paper and the supplementary  material. It is tough to judge the performance of the proposed framework and the baselines without knowing the tasks used to evaluate them. How  easy are the easy tasks? How hard are the hard tasks? How different are  the tasks used for learning and the tasks used for evaluating zero-shot  generalization?
>
> A: The easy tasks involve only one perception primitive and one behavior primitive. An example is as follows.
> ```
> if IsOnFire():
>     PutOutFire()
> ```
> The hard tasks involve concurrently preparing more than one dish and contain at least 5 subroutines. An example is as follows.
> ```
> parallel:
>     1: if is_ordered(“Onion”):
>             Merge(“ChoppedOnion”,”Plate”)
>             Serve(“ChoppedOnion”)
>     2: if is_ordered(“Onion”):
>             Merge(“ChoppedTomato”,”Plate”)
>             Serve(“ChoppedTomato”)
>     3: while (True):
>         If (IsOnFire()):
>               PutOffFire()
> ```
> The tasks used for learning and for evaluating zero-shot generalization are different compositions of the subtasks in the DSL. For example, the hard tasks used for learning involve preparing dish $\textit{SingleOnion}$ and dish $\textit{SingleTomato}$, but for zero-shot generalization test, a new dish $\textit{OnionTomato}$ is added to the evaluation. We have added another section "Examples of Programs Used in Evaluation" in Appendix A.11 to show more task examples.
>
> Q: How to obtain tasks/programs
>
> A: Tasks/programs in a new domain can be devised with program synthesis approaches. Please refer to the "How to scale to new domain" section in the general response for further details.

---

### Author Response · Authors · 2022-08-02
**General Response**

We thank the reviewers for their insightful feedback! We address common concerns here and will reply to each reviewer separately to address the remaining concerns.

### Why assuming full observability

The goal of our work is to solve long-horizon cooperative tasks with rich subtask structures (e.g., video games, multi-drone delivery) where a central controller or inter-agent communication exists. Therefore we follow the original game $\textit{Overcooked}$ and assume full observability.

Despite this, we add an experiment to show that E-MAPP can still outperform other methods in a partially observable environment. In the new setting, the observation of each agent is only part of the map within reach. The results are as follows. Under the new setting, E-MAPP can still learn to allocate sub-tasks to agents and accomplish the tasks efficiently.

| model | score | completion rate |
| :-----: | :-----: | :-----: |
| E-MAPP(original) |  1.58±0.60  |  56.3%    |
| E-Mapp(partial obs) |  1.01±0.38    |   27.1%   |
| MAPPO |  0.59± 0.27    |  0.0%    |

### Computational complexity of subtask allocation

In an environment with $M$ subtasks and $N$ agents, the brute force search for an optimal allocation indeed has a complexity of $O(M^N)$. However, the practical complexity is much smaller than it. The reasons are as follows:
1) In a certain stage of a long-horizon task, only a small amount of subtasks are feasible. Thus,  the subtask amount $M$ can be pruned into a smaller number $L$ by checking the feasibility function $O(M\times N)$ times.
2) The engaging $N$ agents can be classified into $C$ roles. The agents sharing the same role have the same reachability functions. $C$ is often a property of the task that does not scale with $N$. For example, in the overcooked environment, $C$ can be the number of connected components of the map. Note that, in E-MAPP, the assistive agents aim to increase the reachability of the leading agents. We define $C\times L$ new subtasks by pairs $(\tau, c)$, where $\tau$ comes from the $L$ feasible subtasks and $c$ comes from the $C$ roles. The goal of each new subtask $(\tau, c)$ is to help agents with role $c$ to gain reachability on subtask $\tau$. We can obtain a new subtask set of size $O(C\times L)$ by extending the original subtask set with these newly defined subtasks. Assume that the number of agents is smaller than the number of feasible subtasks (otherwise, idle agents will inevitably emerge). Under this assumption, each agent will choose to either complete a subtask alone or assist a certain group of agents with the same role, and each subtask in the new subtask set is allocated to at most one agent to avoid conflict. Then the task allocation problem turns into finding the best matching of $N$ agents and $O(C\times L)$ subtasks with the smallest total cost, which can be solved by the Hungarian algorithm. The  computational complexity is $O((N+CL)^3)\leq O((N+CM)^3)$ that scales well.

We have added an additional experiment with a doubled number of agents to evaluate our algorithm. The results show that E-MAPP can scale to environments with more agents and further boost the time efficiency by parallelization.
|      model       |   score    | completion rate |
| :--------------: | :--------: | :-------------: |
| E-MAPP(original) | 0.99±0.22  |      43.7%      |
|  E-MAPP(larger)  | 1.13± 0.31 |     46.3 %      |

### How to scale to new domain

E-MAPP can also be applied to various domains. When it comes to a new domain, we can devise new perception primitives and behavior primitives based on object properties and interactions among objects. These primitives along with the branching and parallelization keywords compose the DSL. Previous approaches on program synthesis can be applied to synthesize programs for tasks. For example, we can synthesize programs from diverse video demonstrations.  The activities (subtasks) of a task in a video can be segmented out as a subroutine for program extraction. By summarizing the chronological order of subtask completions, we can obtain the dependence of subtasks and put the possibly parallelizable subtasks in one parallel subroutine. We have added this section to Appendix A.14.

---

### Meta-Review · Area_Chair_AtDv · 2022-08-25

**Recommendation:** Accept
**Confidence:** Certain

**Metareview:**

This paper deals with complex long-horizon tasks with multi-agent RL. The authors propose E-MAPP method that leverages parallel programs to guide multiple agents with goals to accomplish the task jointly.
Generally, this paper is with an interesting idea and has sound technical contributions. The presentation is a bonus point of this paper. The rebuttal mostly eases the concerns of the reviewers. As a result, all the reviewers vote for an acceptance of this paper.
The major weakness of the proposed method lies in the inconvenience of applying E-MAPP to a new environment or task since it requires a huge amount of work. Maybe due to this reason, the experiment is conducted on overcooked v2 environment only.
In sum, I think this is an interesting paper tackling a type of challenging task and thus recommend an acceptance of this paper.


**Award:**

No

---

### Decision · Program_Chairs · 2022-09-14

Accept